# FED3+2P: TRAINING DIFFERENT PARTS OF NEURAL NETWORK WITH TWO-PHASE STRATEGY

## ABSTRACT

In federated learning, the non-identically distributed data affects both global and local performance, while clients with small data volumes may also suffer from overfitting issues. To address these challenges, we propose a federated learning framework called Fed3+2p. In Fed3+2p, we divide the client neural network into three parts: a feature extractor, a filter, and classification heads, and to train these parts, we present two types of coordinators to train client sets with a two-phase training strategy. In the first phase, each Type-A coordinator trains the feature extractor of partial clients, whose joint data distribution is similar to the global data distribution. In the second phase, each Type-B coordinator trains the filter and classification heads of partial clients, whose data distributions are similar to each other. We conduct empirical studies on three datasets: FMNIST and CIFAR-10/100, and the results show that Fed3+2p surpasses the state-of-the-art methods in both global and local performance across all tested datasets.

## 1 INTRODUCTION

In Federated Learning, the non-identically distributed nature of data leads to some challenges, such as unstable model parameter aggregation and inefficient communication Alamgir et al. (2022); Kairouz et al. (2021); Tan et al. (2023). Personalized Federated Learning (pFL) has attracted much attention in mitigating feature distribution inconsistency and label distribution imbalance issues in recent years Tan et al. (2023); Dinh et al. (2022); Zhang et al. (2023b;c), and is also the focus of our research.

In pFL, the performance of the global model plays a crucial role in ensuring robustness across diverse datasets, while local optimization allows each client to improve its model based on its specific data, adapting to the unique characteristics of its own dataset Konečný et al. (2015). However, existed pFL approaches pose a problem on existed clients: no or limited improvement of global performance and local performance Zhang et al. (2023a); Hu et al. (2020); Dinh et al. (2022), such as FedPerArivazhagan et al. (2019), FedProx Li et al. (2020b) and pFedMe Dinh et al. (2022). Recently, shared feature extractors combined with personalized classification heads P-Heads Collins et al. (2023); Chen & Chao (2022) or personalized feature extractor Li et al. (2021b); Tan et al. (2022) are proposed to promote global and local performance, respectively. The core idea of this method is to train different parts of the network with proper data samples, so we extend this idea here, by reorganizing the neural network from the "*feature extractor - personalized classification head*" to "*feature extractor - filter - personalized classification head*", and training the three parts separately.

On the one hand, a high-quality feature extractor is essential for these methods, but it will bias the shared feature extractor's parameters away from the global optimum when averagely aggregating Duan et al. (2021). For example, when a client's data dominates the data space or are more concentrated, the feature extractor tends to learn more prominent features from the client, which will cause the feature extractor to reflect the data characteristics of this client more, posing the risk of overfitting on specific client Wang et al. (2020); Karimireddy et al. (2021). Thus, an underperforming feature extractor may mislead the feature extraction process, causing a decline in global performance Chen et al. (2023). In addition, through mathematical proof Duan et al. (2021), we know that when the local data distribution is consistent with the global data distribution, the risk of the model overfitting to specific clients is reduced. Here, to train a high-quality feature extractor, we partition all clients

into several sets whose joint data distributions are similar to the global distribution, and then use coordinators to manage the training of clients.

On the other hand, the personalized classification head P-Head plays a crucial role in the personalized classification ability of each client's local model, which is independently trained on each client's local dataset Sattler et al. (2019); Konečný et al. (2015); Chen & Chao (2022). However, due to the typically simple structure of the P-Head and the lack of external information during independent training, it is prone to overfitting on clients with smaller data volumes Liu et al. (2019); Wang et al. (2020). Here, to address the overfitting issue, we partition all clients into several groups with similar data distributions, designate a part of the neural network as a shared filter part for each client group, and then use coordinators to manage the training of clients.

In corresponding to the three reorganized parts of the neural network, we design a two-phase training strategy. In the first phase, we train the feature extractor shared by all clients. In the second phase, we train the filter shared by clients with similar data distributions, along with the classification heads unique to each client. Overall, we constructed a federated learning framework named Fed3+2p, which features a three-part neural network architecture and a two-phase training strategy, introducing coordinators to manage the client training process.

Our contributions are summarized as follows:

- We propose a federated learning method Fed3+2p, which divides a neural network architecture into three parts: a feature extractor, a filter, and classification heads, and adopts a two-phase training strategy with two types of coordinators managing the clients.

- To train the shared feature extractor of all clients under conditions similar to the global data distribution, we present Type-A coordinators to partition all clients into several sets whose joint data distributions are similar to the global distribution, and then update the parameters of the feature extractor on coordinators and the central server.

- To train the shared filter of some clients under conditions of similar data distributions, we propose Type-B coordinators to cluster clients with similar data distributions, and then update the parameters of the filter on their corresponding coordinator.

## 2 RELATED WORK

In this section, we provide a brief overview of imbalanced data learning and introduce various personalized federated learning algorithms.

### 2.1 IMBALANCED DATA LEARNING

The primary solutions to address this issue involve sampling and ensemble learning Yang et al. (2019); Li et al. (2020a).

Sampling includes undersampling and oversampling. Undersampling involves sampling from imbalanced datasets to obtain a balanced subset, which is easy to implement but requires large datasets.Unfortunately, in FL, the datasets of each client are typically small. Oversampling generates minority class samples to balance the dataset, but generating a large number of duplicate minority class samples on small client datasets may lead to severe overfitting. Ensemble methods are sensitive to noise and outliers, which are common in distributed datasets. While these methods manipulate the dataset to alleviate the non-identically distributed problem, they have limited effectiveness due to aforementioned issues.

### 2.2 PERSONALIZED FEDERATED LEARNING

Existing personalized Federated Learning (pFL) algorithms can be categorized into several main types, each with its unique characteristics.

**Fine-Tuning.** Algorithms in this category, such as Per-FedAvg Fallah et al. (2020), learn a global model through fine-tuning. They perform minor local fine-tuning on each client to adapt to specific client data distributions and features.

**P-Heads.** Including algorithms like FedPer Arivazhagan et al. (2019), FedRep Collins et al. (2023), and FedRoD Chen & Chao (2022), which partition the neural network into a feature extractor and P-Heads. The feature extractor is shared, while each client independently owns P-Heads. This allows for personalized model training based on the specific requirements of each client.

**Regularization Methods.** Algorithms in this category use regularization techniques to balance the relationship between the global model and personalized models, such as FedProx Li et al. (2020b), pFedMe Dinh et al. (2022), and Ditto Li et al. (2021a). They introduce additional regularization terms to control the distance between the global model and personalized models, achieving a balance between global performance and personalized performance.

An ideal pFL is a type of FL with two objectives: (1) aggregating information for collaborative learning and (2) training reasonable personalized models Zhang et al. (2023a). However, existing pFL methods often focus on only one of these objectives on clients. FedPer Arivazhagan et al. (2019) and FedRep Collins et al. (2023) rely solely on local data for personalized training, lacking the learning of global knowledge, which makes them prone to overfitting when client data is limited. FedRoD Chen & Chao (2022) trains the feature extractor to capture global features but overlooks personalized feature extraction.

# 3 OBJECTIVES OF PERSONALIZED FEDERATED LEARNING

The personalized federated learning has a global optimization objective and a set of local objectives. The local objectives focus on optimizing for each client, while the global objective aims to find the best global model across all clients.

Let $N$ be the number of clients, and $D^{(k)}$ be the local dataset on client $k$, we denote the expected loss on client $k$ as $F^{(k)}$, shown by Eq. (1)

$$F^{(k)} = \mathbb{E}_{(\mathbf{x},y)\sim D^{(k)}}[L(f(\mathbf{x};W^{(k)}),y)], \tag{1}$$

where $W^{(k)}$ denotes the parameters of the entire neural network on client $k$, $f(\cdot)$ denotes the model's prediction function, $L(\cdot)$ denotes the loss function applied to each local data instance, and $\mathbb{E}$ denotes the expectation operator. And so the local objective of client $k$ is defined by Eq. (2)

$$\min F^{(k)}, \quad k = 1, 2, \ldots, N. \tag{2}$$

Furthermore, the global objective can be expressed as the sum of weighted local objectives, shown by Eq. (3)

$$\min \sum_{k=1}^{N} \alpha_k F^{(k)}, \tag{3}$$

where the weight of client $k$ is $\alpha_k = \frac{|D^{(k)}|}{\sum_{j\in N}|D^{(j)}|}$, and $|D^{(k)}|$ is the number of training samples on client $k$.

From Eq. (2) and Eq. (3), the challenge of achieving both local and global optimization objectives lies in their differing focuses: while the local objective minimizes loss on each client's local data distribution $D^{(k)}$, potentially reducing model parameter coordination across clients, the global objective balances personalized objectives $F^{(k)}$ through weighted summation, which can weaken individual client model personalization.

# 4 METHOD

Our proposed federated learning framework, namely Fed3+2p, is structured into three layers: the central server, the Type-A/B coordinator, and the clients. The training of this framework is divided into two phases.

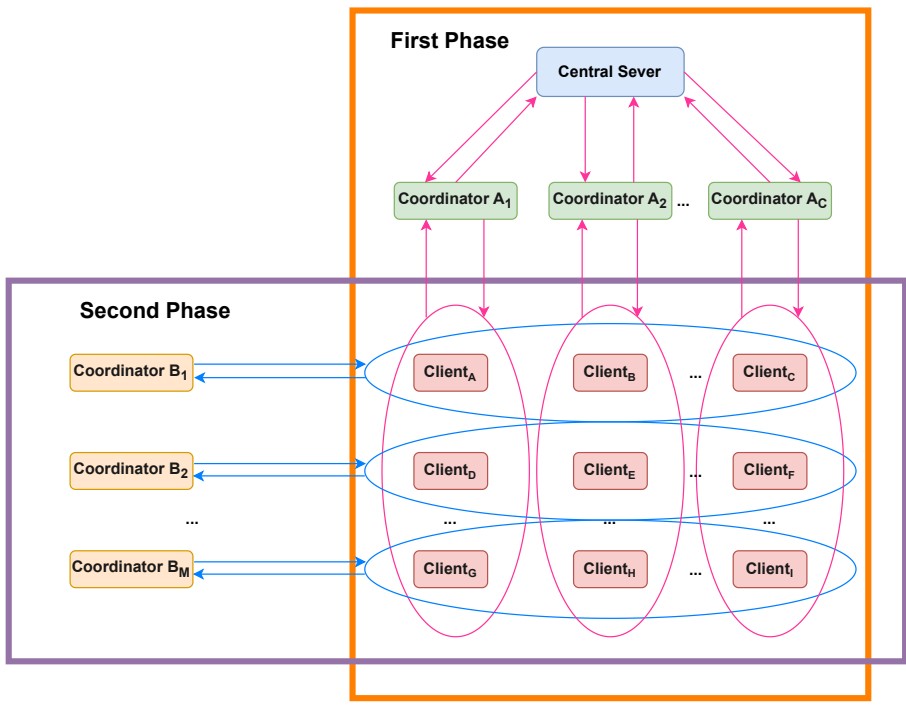

Figure 1. Illustration of the structure of the federated learning framework Fed3+2p. The central server is responsible for collecting information sent by the clients, and performing aggregation tasks. The Type-A coordinator manages clients whose overall data distribution is similar to the global data distribution, and is responsible for collecting gradient updates from the clients it manages, uploading them to the central server, and distributing the aggregated global parameters from the central server to each client during the first phase. The Type-B coordinator manages clients with similar data distributions, and is responsible for collecting gradient updates from the clients it manages, aggregating them, and then distributing the updates to each client during the second phase.

## 4.1 FIRST PHASE

Our goal is to train a feature extractor shared by all clients in this phase. The central server, coordinators, and clients all participate in the training process.

**Central Sever.** The central server is primarily responsible for aggregating the parameters from the coordinators. Given the global model parameters $W(t)$ in round $t$ and the model parameter updates $\Delta W^c(t+1)$ computed by coordinator $c$ in round $t$, we update the global model parameters $W(t+1)$ by Eq. (4)

$$W(t + 1) = W(t) - \frac{1}{|D|} \sum_{c=1}^{C} |D^c| \Delta W^c(t + 1), \tag{4}$$

where $C$ is the total number of coordinators, $|D^c|$ is the total number of training samples handled by coordinator $c$, $D = \bigcup D^c$ is all training samples and its cardinality is $|D|$. Subsequently, the central server sends the aggregated parameters $W(t + 1)$ to each coordinator.

At the end of this phase, $W$ will be saved as the global model parameters.

**Type-A Coordinators.** All clients are grouped and managed by some coordinators. Each client is managed by exactly one coordinator. The joint data distribution of the clients managed by a coordinator is regarded as the coordinator's data distribution. All clients managed by the same coordinator ultimately share the same parameters.

First, to group all clients while ensuring the consistency between the coordinator's data distribution and the global data distribution, we use KL divergence as the grouping metric, shown in Eq. (5)

$$\min \sum_{c=1}^{C} \text{KL}\left(P_{D^c} \,\|\, P_D\right), \tag{5}$$

where $P_{D^c}$ represents the label probability distribution of coordinator $c$, and $P_D$ denotes the label probability distribution of the global data. We approximate these probability distributions using empirical distributions.

Second, the coordinator manages all clients within its group to update its parameters sequentially. This process consists of three steps:

*Step 1*: At the beginning of round $t$ of training, coordinator $c$ receives the parameters $W(t)$ from the central server and sends them to a randomly selected client $k$ that it is responsible for, as shown in Eq. (6)

$$W^{(k)}(t) = W^c(t) = W(t), \tag{6}$$

where $W^{(k)}(t), W^c(t)$ are the model parameters of client $k$ and coordinator $c$, respectively, and their initial values are same.

*Step 2*: The coordinator receives the parameter update $\Delta W^{(k)}$ from client $k$ and then updates $W^c(t)$ by Eq. (7)

$$W^c(t) = W^c(t) - \Delta W^{(k)}. \tag{7}$$

Then, the updated $W^c(t)$ is sent to a client that has not yet undergone local training.

*Step 3*: After all the clients managed by coordinator $c$ have completed local training, the total parameter update $\Delta W^c(t+1)$ is calculated by Eq. (8)

$$\Delta W^c(t+1) = W(t) - W^c(t). \tag{8}$$

The total parameter update $\Delta W^c(t+1)$ is then sent to the central server.

In addition, to reduce additional communication overhead and hardware resource consumption, the coordinators are deployed directly on the central server. Therefore, the computational overhead of the coordinators is equivalent to the computational overhead on the central server.

**Clients.** Our client network contains three parts: a feature extractor, a filter, and G-/P-Heads, illustrated by Figure 2. Accordingly, all trainable parameters of our client network are denoted by $W := \{W_e, W_i, W_{g/p}\}$.

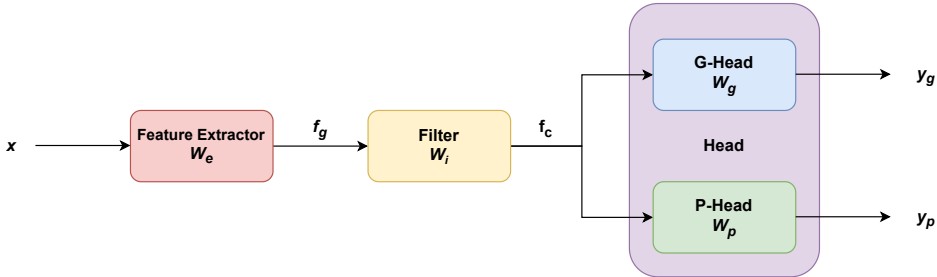

Figure 2. Illustration of the modules in the client neural network and the data flow between them. We first use the feature extractor to transform the raw input $x$ into a feature vector $f_g$, then input $f_g$ into the filter to obtain the feature vector $f_c$, which is adapted to the set of similar clients $S_c$. Finally, $f_c$ is input into the G-Head and the P-Head to produce the global prediction result $y_g$ and the local prediction result $y_p$. We share the parameters $W_e$ across all clients and share the parameters $W_i^c$ among the clients in the client set $S_c$ managed by coordinator $c$.

For a given client $k$ managed by coordinator $c$, its initial parameters $W^{(k)}(0) = W(0)$ are released by the central server and its parameters are calculated by Eq. (9)

$$\Delta W^{(k)} \leftarrow \eta \nabla_{W^{(k)}(t)} L, \tag{9}$$

where $W^{(k)}(t)$ contains all trainable parameters $W_e, W_i$ and $W_g$, thereafter, the parameter updates $\Delta W^{(k)}$ are uploaded to coordinator $c$.

## 4.2 SECOND PHASE

Our goal is to train a filter shared by clients with similar data distributions, as well as personalized heads for each client in this phase. Only the coordinators and clients participate in the training process.

**Type-B Coordinators.** The Type-B coordinator is responsible for grouping clients and aggregating their parameter updates. The coordinator no longer communicates with the central server. Clients managed by the same coordinator share the parameters of the filter.

First, to group all clients while ensuring the similar label distributions on clients within the same coordinator, we use JS divergence as the grouping metric, shown in Eq. (10)

$$\{S_1^*, S_2^*, \ldots, S_M^*\} = \underset{S_1, S_2, \ldots, S_M}{\arg\min} \sum_{c=1}^{M} \sum_{i,j \in S_c} \text{JS}(P_{D^{(i)}} \parallel P_{D^{(j)}}), \tag{10}$$

where $P_{D^{(i)}}$ represents the label probability distribution of client $i$, $M$ denotes the total number of coordinators, and $S_c^*$ is the set of clients managed by coordinator $c$.

And then, coordinator $c$ aggregates all updates from clients by Eq. (11)

$$W^c(t+1) = W^c(t) - \frac{1}{|D^c|} \sum_{k \in S_c^*} |D^{(k)}| \Delta W^{(k)}, \tag{11}$$

where $W^c(t)$ represents the model parameters of coordinator $c$ updated in round $t$ and $\Delta W^{(k)}$ denotes the parameter update of client $k$.

**Clients.** The filter and the P-Head parameters $\{W_i, W_p\}$ of all clients are randomly reinitialized, while their feature extractor and G-Head parameters $\{W_e, W_g\}$ are initialized as the outputs of the first training phrase and are frozen and remain unchanged during the second phrase.

In detail, for any client $k$, the updates of its filter parameters $W_i^{(k)}$ and P-head parameters $W_p^{(k)}$ are respectively calculated by Eq. (12) and (13)

$$\Delta W^{(k)} \leftarrow \eta \nabla_{W_i^{(k)}(t)} L, \tag{12}$$

$$W_p^{(k)}(t+1) = W_p^{(k)}(t) - \eta \nabla_{W_p^{(k)}(t)} L, \tag{13}$$

where $\Delta W^{(k)} = \{0, \Delta W_i^{(k)}, 0\}$, thereafter, the parameter updates $\Delta W^{(k)}$ are uploaded to coordinator $c$. The P-Head only participates in local training and does not participate in aggregation in the second phase.

## 5 EXPERIMENT

To verify the global and local performance of Fed3+2p, we compare it with the SOTA methods on widely used datasets: FMNIST and CIFAR-10/100. Additionally, to assess the effectiveness of the coordinator, filter, and two-phase training method in Fed3+2p, we conduct ablation studies.

### 5.1 SET UP

**Datasets.** We evaluate our proposed Fed3+2p on the following three datasets: FMNIST Xiao et al. (2017) and CIFAR-10/100 Krizhevsky et al. (2009). The method for splitting the training set and test set is also consistent with that of Xiao et al. (2017) and Krizhevsky et al. (2009).

**Backbones.** Based on the work of McMahan et al. (2023); Chen & Chao (2022); Acar et al. (2021), we use a ConvNet LeCun et al. (1998). For the FMNIST datasets, the model consists of 2 convolutional layers and 2 fully connected layers. The convolutional layers have 32 and 64 channels, respectively. The fully connected layers have 50 neurons as the hidden size and 10 neurons as the output for 10 classes. For the CIFAR-10/100 datasets, the model consists of 3 convolutional layers and 2 fully connected layers. The convolutional layers have 32, 64, and 64 channels, respectively. The fully connected layers have 64 neurons as the hidden size and 10/100 neurons as the output for 10/100 classes.

**Statistically heterogeneous settings.** To simulate non-identically distributed data distribution on CIFAR-10/100 and FMNIST, we create heterogeneous data partitions for $K$ clients following work Hsu et al. (2019). Suppose the dataset has $R$ classes. For each class $r$, a $K$-dimensional probability vector $q_r$ is generated from the Dirichlet distribution $\text{Dir}(\alpha)$, where $q_r = [q_r[1], q_r[2], \ldots, q_r[K]]$. Each element $q_r[k]$ of this vector represents the probability of assigning data from class $r$ to client $k$. The data of each class $r$ is allocated to client $k$ according to the probability $q_r[k]$. Specifically, if there are $N_r$ samples in class $r$, then client $k$ will receive approximately $q_r[k] \times N_r$ samples. Since $\alpha < 1$, the generated probability vectors $q_r$ are typically very imbalanced, meaning that the data for some classes may be concentrated on a few clients, while other clients may have fewer or none of these class samples.

**Baselines.** Specifically, we compare Fed3+2p with twelve federated learning algorithms, including FedAvg McMahan et al. (2023), FedProx Li et al. (2020b), Per-FedAvg Fallah et al. (2020), pFedMe Dinh et al. (2022), Ditto Li et al. (2021a), FedPer Arivazhagan et al. (2019), FedRep Collins et al. (2023), FedRoD Chen & Chao (2022), FedDYN Acar et al. (2021), CFL Sattler et al. (2019), PACFL Vahidian et al. (2022), and FedClust Islam et al. (2024). Additionally, we conduct ablation experiments to demonstrate the effectiveness of each module.

**Metric.** To evaluate global performance, we test the global model (GM) on the global test set (G-Test) and also on the local test set (P-Test) using the global model (GM). To assess local performance, we test the local model (LM) on the local test set (P-Test). For each local test set, its data distribution is consistent with each local training set.

**Hyperparameters.** We conduct 100 training rounds for each method. Consistent with Li et al. (2020b), the local learning rate is set to decay over communication rounds. For FMNIST, the initial learning rate is set to 0.01, and for CIFAR-10/100, the initial learning rate is set to 0.001, which is reduced by a factor of 0.99 in each round, similar to the approach in Acar et al. (2021). We reset the learning rate at the beginning of each training phase in Fed3+2p. We use M = 100 clients for FMNIST and M = 20 for CIFAR-10/100, and sample 20%/40% clients at every round. For the duration of the experiments, we utilize the SGD optimizer with a weight decay of 1e-5 and a momentum of 0.9. The mini-batch size is 40. During each round, clients perform local training over 5 epochs. The results presented are the mean values from five separate experiments conducted with different random seeds.

## 5.2 Comparison with State-Of-The-Art

The experimental results of our method compared to other methods are shown in Table 1. As shown in Table 1, the second and third rows indicate the type of model used and the test dataset the evaluation conducted on, respectively. The first column, from the fourth row to the last row, contains the names of the SOTA methods as well as our method, Fed3+2p. The second to ninth columns contain the test accuracy results on the FMNIST, Cifar10, and Cifar100 datasets, with specific models evaluated on specific test datasets. From Table 1, we draw three conclusions.

First, our method, Fed3+2p, achieved significant improvements on three test datasets in global performance compared to all baseline federated learning methods. Compared to the best personalized federated learning methods, Fed3+2p outperformed FedRoD in terms of accuracy on both the global and local test sets of FMNIST by 3.6% and 3.5%, respectively, and improved by 12.9% and 15.2% on both the global and local test sets of Cifar10. For Cifar100, its global performance is nearly on par with the best personalized federated learning methods. Even when compared to the best general federated learning method, Fed3+2p surpassed FedDYN on the global and local test sets of FMNIST by 4.3% and 4.2%, respectively; on Cifar10, it improved by 18.0% and 19.8%, and on both test sets of Cifar100, the improvement was 2.9% and 2.3%, respectively.

Table 1: Global accuracy and local accuracy comparisons of different approaches over different datasets for non-IID Dir (0.1) (%). Model Type represents the type of model used for testing, distinguishing between global model GM and local models LM. Test Set refers to the type of test set, differentiating between global test sets G-Test and local test sets P-Test. The **bold** numbers represent the best results for each model on each test set for every dataset. Red font indicates general federated learning methods, blue font indicates clustered federated learning methods, and cyan font indicates personalized federated learning methods. † indicates results collected from papers, while ‡ indicates results obtained from our re-implemented code.

| Dataset | FMNIST | | | Cifar10 | | | Cifar100 | | |
|---|---|---|---|---|---|---|---|---|---|
| Model Type | GM | | LM | GM | | LM | GM | | LM |
| Method/Test Set | G-Test | P-Test | P-Test | G-Test | P-Test | P-Test | G-Test | P-Test | P-Test |
| Local only‡ | - | - | 85.9 | - | - | 87.4 | - | - | 40.2 |
| FedAvg‡ | 81.1 | 81.3 | 91.5 | 57.9 | 57.8 | 90.6 | 41.8 | 41.8 | 70.2 |
| FedDYN† | 83.2 | 83.2 | 90.7 | 63.4 | 63.9 | 92.4 | 43.0 | 43.0 | 72.0 |
| FedProx† | 82.2 | 82.3 | 91.4 | 58.7 | 58.9 | 89.7 | 41.7 | 41.6 | 70.4 |
| CFL† | - | - | 75.2 | - | - | 41.9 | - | - | 33.4 |
| PACFL† | - | - | 85.6 | - | - | 51.3 | - | - | 47.8 |
| FedClust† | - | - | **95.3** | - | - | 59.7 | - | - | 49.5 |
| Ditto† | 81.5 | 81.5 | 89.4 | 58.1 | 58.3 | 86.8 | 41.7 | 41.8 | 68.5 |
| FedPer† | 74.5 | 74.4 | 91.3 | 50.4 | 50.2 | 89.9 | 37.6 | 37.6 | 71.0 |
| FedRep† | 79.5 | 80.1 | 91.8 | 56.6 | 56.2 | 91.0 | 40.7 | 40.7 | 71.5 |
| FedRoD† | 83.9 | 83.9 | 92.7 | 68.5 | 68.5 | 92.7 | **45.9** | **45.8** | 72.2 |
| Per-FedAvg† | 80.5 | - | 82.8 | 60.7 | - | 82.7 | 39.0 | - | 66.6 |
| pFedMe† | 76.7 | 76.7 | 83.4 | 50.6 | 50.7 | 76.6 | 38.6 | 38.5 | 63.0 |
| Fed3+2p(Ours) | **87.5** | **87.4** | 91.9 | **81.4** | **83.7** | **94.1** | **45.9** | 45.3 | **72.6** |

Secondly, in terms of local performance, our method Fed3+2p surpasses all baseline federated learning methods on the CIFAR-10/100 datasets. Specifically, compared to the best general federated learning method, Fed3+2p outperforms FedDYN by 1.7% and 0.6% on CIFAR-10 and CIFAR-100, respectively. When compared to the best clustered federated learning method, Fed3+2p exceeds FedClust by 34.4% and 23.1% on CIFAR-10 and CIFAR-100, respectively. Even when compared to the best personalized federated learning method, Fed3+2p achieves 1.4% and 0.6% higher accuracy than FedRoD on CIFAR-10 and CIFAR-100. However, Fed3+2p does not perform as effectively as FedClust on the FMNIST dataset.

Finally, Fed3+2p demonstrates a clear advantage on more complex datasets. In the FMNIST dataset, due to its lower complexity and relatively simple image features, the advantage of Fed3+2p is relatively small, with global performance only 3.6% higher than the best SOTA method, and almost no advantage in local performance. However, in the CIFAR-10 dataset, the advantage of Fed3+2p over SOTA methods is extremely pronounced, with global and local performance exceeding that of the best SOTA method by 12.9% and 1.4%, respectively. On the other hand, in CIFAR-100, despite the excellent structural design of Fed3+2p, the neural network used in the experiments is too simple to effectively capture the complex features of the dataset, resulting in almost no advantage compared to SOTA methods. Therefore, future research could consider more complex network architectures to better accommodate high-complexity datasets like CIFAR-100.

To gain deeper insights into our Fed3+2p and identify the reasons behind these improvements, we further investigate our training method and network architecture in the following empirical studies.

## 5.3 ABLATION EXPERIMENTS

The ablation experiments provide an in-depth effectiveness analysis of the filter, coordinator, and two-phase training strategy of our Fed3+2p.

In the ablation experiments, we adopted a progressive removal strategy, rather than independently removing the filter, coordinator, and two-phase training strategy. Specifically, when removing the

filter, we turn off the Type-B coordinator in the second phase but retained the management functionality of the Type-A coordinator. Then, in the coordinator removal experiment, we turn off both the Type-A and Type-B coordinators, while still maintaining the phased training of the feature extractor and other components. Finally, when removing the two-phase training strategy, the entire network was trained together without the use of any coordinators for client management. The results of the ablation experiments are shown in Table 2.

Table 2: The accuracy (%) of our ablation experiments on Fed3+2p. Bold numbers represent the best results. 2ST. Train denotes the two-phase training method. **G** represents testing the global test set with the global model, while **L** represents testing local data with the local model.

| Method | FMNIST | | Cifar10 | | Cifar100 | |
|---|---|---|---|---|---|---|
| | **G** | **L** | **G** | **L** | **G** | **L** |
| Full Fed3+2p | **87.5** | **91.9** | **81.4** | **94.1** | **45.9** | **72.6** |
| - Filter | - | 91.6 | - | 93.7 | - | 72.5 |
| - Coordinator | 82.7 | 89.8 | 64.0 | 91.7 | 40.7 | 65.6 |
| - 2ST. Train | 82.7 | 89.2 | 63.9 | 90.5 | 40.7 | 64.2 |

**Effectiveness of Filter**: In the first phase of the ablation experiments, removing the filter leds to a decline in local performance. In the FMNIST dataset, local performance decreased slightly by 0.3%, while in CIFAR-10, it dropped by 0.4%. In CIFAR-100, there was almost no change, with only a 0.1% decrease. These results show that the filter enhances the model's local performance, especially on simpler datasets like FMNIST and CIFAR-10, where it helps capture local data features and mitigate overfitting issues that may arise in clients with smaller datasets. However, on more complex datasets like CIFAR-100, the filter has relatively less impact, with the model relying more on other components to improve performance.

**Effectiveness of Coordinator**: In the second phase of the ablation experiments, removing both the Type-A and Type-B coordinators results in a significant decline in global and local performance. In the FMNIST dataset, global performance dropped by 4.8% and local performance by 2.1%. In the CIFAR-10 dataset, global performance decreased by 17.4% and local performance by 2.4%. In the CIFAR-100 dataset, global performance fell by 5.2% and local performance by 7.0%. These results demonstrate that the coordinators enhance both global and local performance. Specifically, the Type-A coordinator mitigates model bias caused by non-identically distributed client data by grouping clients into sets that are similar in joint data distribution to the global data distribution and managing their training, which allows each group's model updates to reflect a more comprehensive global knowledge, thereby enhancing global performance. Similarly, the feature extractor also learns richer feature representations in this process, which can be directly utilized during the training of the personalized parts in the second phase, allowing them to better adapt to their respective data distributions and improve local performance.

**Two-phase Strategy**: In the third phase of the ablation experiments, we remove the two-phase training strategy and disable all coordinators, resulting in a further decline in local performance. In the FMNIST dataset, local performance decreased by 2.7%; in the CIFAR-10 dataset, local performance dropped by 3.6%; and in the CIFAR-100 dataset, local performance fell by 8.4%. These results indicate that the two-phase training strategy can enhance the model's local performance. By separating the training of the feature extractor from that of the classification head, the feature extractor can learn more comprehensive global features in the first phase, thereby providing more effective feature representations for the classification head, especially when local data distributions are diverse.

Through progressive ablation experiments, we demonstrated the cumulative impact of the filter, coordinator, and two-phase training strategy on the model's global and local performance. The experimental results indicate that the filter primarily affects the model's local performance as expected, while the coordinator enhances the performance of both the global and local models. The two-phase training strategy further improves both global and local performance by facilitating staged training. Therefore, maintaining the synergy among these modules is key to the success of the Fed3+2p framework.

## 5.4 Visualize Feature Vectors

The purpose of visualizing feature vectors is to intuitively demonstrate the important roles of co-ordinators and filters in feature learning. By visualizing the feature vectors from different training phases, we aim to validate their effectiveness in enhancing feature extraction and reducing overfitting. We conducted experiments on Cifar10 under the Dir (0.1) setting while keeping other settings constant.

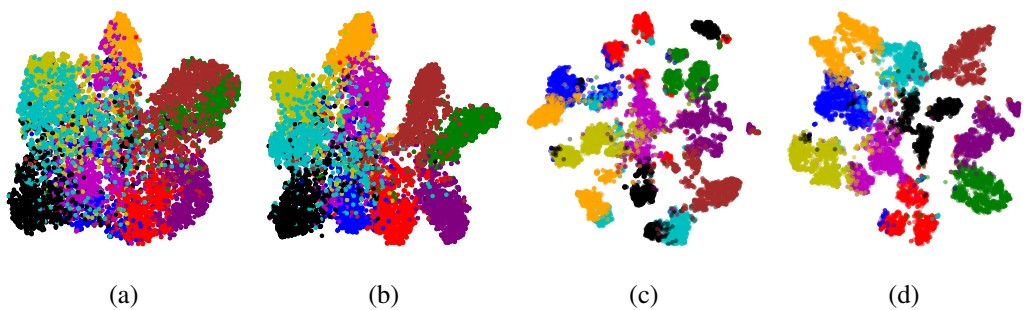

|        |        |        |        |
|:------:|:------:|:------:|:------:|
| (a)    | (b)    | (c)    | (d)    |

Figure 3. Each color represents the feature vectors of one class of data. (a): t-SNE visualization of feature vectors for each class of data without using Type-A coordinators in the first phase. (b): t-SNE visualization of feature vectors for each class of data with using Type-A coordinators in the first phase. (c): t-SNE visualizations of feature vectors for each class of data on each client without using Type-B coordinators in the second phase. (d): t-SNE visualizations of feature vectors for each class of data on each client with using Type-B coordinators in the second phase.

As shown in Figure 3, we used t-SNE to visualize the feature vectors of each class of data obtained from client training managed with and without a coordinator during two training phases.

In Figure 3a, the feature vectors of different classes exhibit a noticeable clustering phenomenon, while in Figure 3b, the feature vectors of different classes show good separability. This indicates that managing the training process of clients using Type-A coordinator helps the feature extractor learn the feature differences between different categories, thereby enhancing the performance of the feature extractor.

In Figure 3c, the feature vectors of the same label demonstrate considerable dispersion; for instance, the black feature vectors are partially located in the upper right and partially in the lower center. In contrast, Figure 3d shows a significant aggregation of the feature vectors with the same label, indicating that using Type-B coordinator to manage clients with similar data distributions, in combination with shared filters and feature extractors, can better capture the shared features of the same labels across different clients, reducing the overfitting phenomenon on specific client data.

## 6 Conclusion and Future Work

In this paper, we propose a novel federated learning framework called Fed3+2p to address the challenges posed by overfitting issues faced by non-identically distributed data and clients with small data volumes, affecting both global and local performance. Fed3+2p divides the neural network into three parts: a feature extractor, a filter, and classification heads, and introduces two types of coordinators to implement a two-phase training strategy for these components. Experimental results demonstrate that Fed3+2p significantly outperforms all SOTA methods in both global and local performance.

In future research, we aim to further enhance the flexibility and adaptability of the Fed3+2p framework. One potential improvement is to establish a threshold mechanism for automatically switching between the two-phase training process, which would autonomously decide when to use Type-A or Type-B coordinators. This would make the training process more efficient and better suited to different types of clients. Additionally, we plan to explore using other metrics to group clients, enhancing privacy by collecting the minimum necessary information from clients.

REPRODUCIBILITY STATEMENT

We provide detailed information about the hyperparameters, datasets, evaluation, and other details in Section 5, which should be comprehensive enough for reproducibility of the experiments. The source code related to the paper is uploaded as supplementary material.

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

## A ADDITIONAL EXPERIMENTS AND ANALYSES

### A.1 ADDITIONAL COMPARISON WITH STATE-OF-THE-ART

Due to space limitations in the main paper, we provide comparisons of global accuracy and local accuracy of different approaches under other non-independent and identically distributed settings across different datasets here, as detailed in Table 3.

Table 3: Global accuracy and local accuracy comparisons of different approaches over different datasets for non-IID Dir (0.3) (%). Model Type represents the type of model used for testing, distinguishing between global model GM and local models LM. Test Set refers to the type of test set, differentiating between global test sets G-Test and local test sets P-Test. The **bold** numbers represent the best results for each model on each test set for every dataset. Red font indicates general federated learning methods, and cyan font indicates personalized federated learning methods. † indicates results collected from papers, while ‡ indicates results obtained from our re-implemented code.

| Dataset | FMNIST | | | Cifar10 | | | Cifar100 | | |
|---|---|---|---|---|---|---|---|---|---|
| Model Type | GM | | LM | GM | | LM | GM | | LM |
| Method/Test Set | G-Test | P-Test | P-Test | G-Test | P-Test | P-Test | G-Test | P-Test | P-Test |
| Local only‡ | - | - | 85.2 | - | - | 75.8 | - | - | 32.5 |
| FedAvg‡ | 83.5 | 83.4 | 90.5 | 68.8 | 69.4 | 85.1 | 46.4 | 46.3 | 61.7 |
| FedDYN† | 86.1 | 86.1 | 91.5 | 72.5 | 73.2 | 85.4 | 47.5 | 47.4 | 62.5 |
| FedProx† | 84.5 | 84.5 | 89.7 | 69.9 | 69.8 | 84.7 | 46.5 | 46.4 | 61.5 |
| Ditto† | 83.3 | 83.2 | 90.1 | 69.7 | 69.8 | 81.5 | 46.4 | 46.4 | 58.8 |
| FedPer† | 79.9 | 79.9 | 90.4 | 64.4 | 64.5 | 84.9 | 40.3 | 40.1 | 62.5 |
| FedRep† | 80.6 | 80.5 | 90.5 | 67.7 | 67.5 | 85.2 | 46.0 | 46.0 | 62.1 |
| FedRoD† | 86.3 | 86.3 | 94.5 | 76.9 | 76.8 | 86.4 | **48.5** | **48.5** | 62.3 |
| Per-FedAvg† | 84.1 | - | 86.7 | 70.5 | - | 80.7 | 44.5 | - | 58.9 |
| pFedMe† | 79.0 | 79.0 | 83.4 | 62.1 | 61.7 | 70.5 | 41.4 | 41.1 | 53.4 |
| Fed3+2p(Ours) | **87.8** | **87.7** | 91.6 | **81.7** | **81.9** | **88.4** | 48.3 | **48.5** | **64.0** |

### A.2 HYPERPARAMETER EXPERIMENT

The objective of this experiment is to explore the impact of the number of Type-B coordinators on local performance in Fed3+2p. We aim to investigate how adjusting the number of Type-B coordinators affects local performance on the CIFAR-10 dataset under the non-IID Dir(0.1) and Dir(0.3) setting.

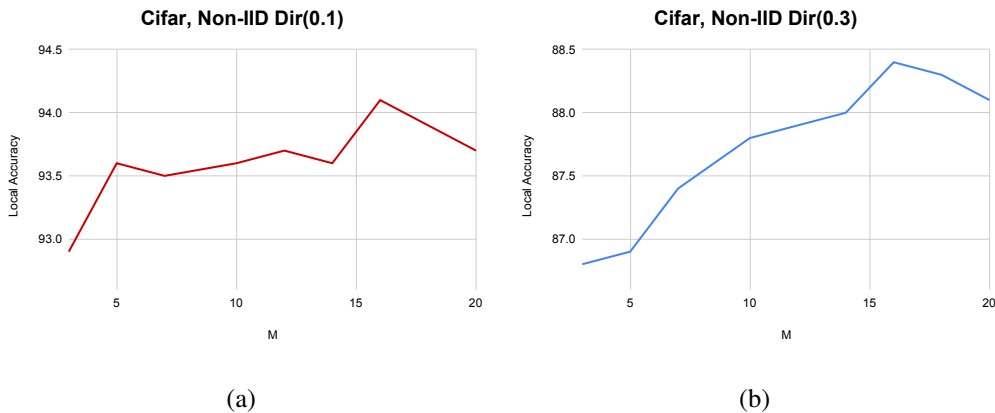

(a)  (b)

Figure 4. Comparison of local accuracy for different numbers of Type-B coordinators on the CIFAR dataset under non-IID Dir (0.1) and Dir (0.3) (%). M represents the numbers of Type-B coordinators.