# OpenReview forum: "Fed3+2p: Training different parts of neural network with two-phase strategy"
_ICLR.cc/2025/Conference — ICLR 2025 Conference Withdrawn Submission_

### Official Review · Reviewer_RdTV · 2024-10-20

**Soundness:** 3
**Presentation:** 3
**Contribution:** 2
**Rating:** 5
**Confidence:** 4

**Summary:**

This paper presents a two-stage training scheme for personalized federated learning, using coordinators to manage client groupings. While the paper is well-presented with thorough experiments, there are concerns about novelty and areas requiring further clarification.

**Strengths:**

1. The paper is clearly written and well-structured.
2. The experimental evaluation is comprehensive, comparing multiple baselines across different datasets.

**Weaknesses:**

1. **The novelty of the approach is not sufficiently articulated.** It's unclear whether: (a) The use of coordinators for client grouping is novel. Have similar dataset/client splitting approaches been tried before? (b) The two-phase training scheme is not novel as it is frequently seen in personalized federated learning (pFL). In terms of progressive training, the idea has been explored in "FedBug: A Bottom-Up Gradual Unfreezing Framework for Federated Learning"), and a direct comparison would be helpful to better understand the novelty.

2. **Empirical validation.** (a) The paper lacks experiments in IID scenarios, which would help demonstrate the method's robustness across different data distributions. (b) The learning dynamics (e.g., train/test loss profiles) may be helpful to understand the method's behavior better.

**Questions:**

1. Why are different divergence measures used in different stages (KL divergence in the first phase, JS divergence in the second)? Theoretical or empirical validation for these choices would strengthen the paper.
2. Could the coordinator setup introduce any privacy concerns?
3. Can FED3+2P be combined with other federated learning techniques like FedDyn, FedProx, or FedExp? This could potentially enhance its performance further.
4. Can the authors clarify the statement "we turn off the Type-B coordinator in the second phase but retained the management functionality of the Type-A coordinator". Does this mean there is no second phase in this ablation? Also, additional experiments with random Type-B assignment would be informative.

---

> ### Author Response · Authors · 2024-11-21
> **Rebuttal by Authors**
>
> Thank you for your valuable feedback. The following is our responses to the questions you raised.
>
> Q1: Why are different divergence measures used in different stages (KL divergence in the first phase, JS divergence in the second)? Theoretical or empirical validation for these choices would strengthen the paper.
>
> A1: The decision to use different divergence metrics at different stages is based on the specific requirements of each phase.
>
> In the first phase, we only need to compare the aggregated data distribution of clients managed by different coordinators with the same global data distribution. For this purpose, the asymmetric KL divergence is appropriate. In the second phase, however, we need to compare the data distributions of clients managed by the same coordinator pairwise, which necessitates a symmetric measure. The JS divergence, as the symmetric form of KL divergence, meets this requirement perfectly.
>
> Moreover, using JS divergence for client clustering is one of the innovations of our work. We hope this explanation clarifies our rationale and demonstrates the thoughtful design behind our methodology.
>
> Q2: Could the coordinator setup introduce any privacy concerns?
>
> A2: In the Fed3+2p framework, coordinators are required to classify clients, and this classification is based on the label distribution information of the clients. We acknowledge that label distribution could potentially be exploited in attacks. However, to date, we have not encountered any concrete methods that can infer original data solely from label distribution information. Furthermore, similar approaches using label distribution information can be found in the following references:
>
> [1] Tao Sheng, Chengchao Shen, Yuan Liu, Yeyu Ou, Zhe Qu, and Jianxin Wang. Modeling global distribution for federated learning with label distribution skew, 2022. URL: https://arxiv.org/abs/2212.08883
>
> [2] Jun Luo and Shandong Wu. FedSLD: Federated learning with shared label distribution for medical image classification, 2021. URL: https://arxiv.org/abs/2110.08378.
>
> [3] A. Khalil et al., "Label-Aware Aggregation for Improved Federated Learning," 2023 Eighth International Conference on Fog and Mobile Edge Computing (FMEC), Tartu, Estonia, 2023, pp. 216-223, doi: 10.1109/FMEC59375.2023.10306055.
>
> [4] Zhu Z, Hong J, Zhou J. Data-free knowledge distillation for heterogeneous federated learning[C]//International conference on machine learning. PMLR, 2021: 12878-12889.
>
> Q3: Can FED3+2P be combined with other federated learning techniques like FedDyn, FedProx, or FedExp? This could potentially enhance its performance further.
>
> A3: Thank you for your suggestion. Based on the same experimental settings described in our paper, we conducted additional experiments to combine our method, Fed3+2p, with FedDyn, FedProx, and FedExp to explore whether these combinations could further enhance performance. The experimental results are as follows:
>
> |     Dataset     | |          | |  Cifar10    | |     | |     | |  Cifar100      | | |
> |------|-|-------|-|-------|-|--------|-|-------|-|-------|-|-------|
> |     Model Type     | |          |GM |          | | LM    | |     |GM  |       || LM |
> |     Method/Test Set     ||      G-Test     | |   P-Test      ||  P-Test    | |  G-Test   ||  P-Test       || P-Test |
> |     Fed3+2p + FedExp     |  |   81.4		   | |    83.3	     | | 94.0	    |  |45.9 | |    45.4   | |72.4    |
> |     Fed3+2p + FedProx     |  |    81.3	   | |    81.5     | | 91.9	    |  |45.5	    | |    45.8    | |67.3    |
> |     Fed3+2p + FedDyn     |  |    81.6	   | |    81.4	     | | 93.5	    |  |46.1    | |  45.7    | |69.2    |
> |      Fed3+2p    |  |   81.4    | |   83.7	      ||   94.1	   | |  45.9   |  |  45.3	      || 72.6    |
>
> The results indicate the following:
>
> Fed3+2p + FedExp: Performance remains almost unchanged. This may be because FedExp focuses on accelerating convergence, with its impact mainly reflected in reducing the number of training rounds rather than directly improving final accuracy.
>
> Fed3+2p + FedProx: The global model (GM) performance is nearly unchanged, but the local model (LM) performance declines. This could be attributed to FedProx’s proximity term, which penalizes the deviation of client models from the global model. This constraint might inhibit the ability of local models in the Fed3+2p framework to adapt to client-specific data, thus reducing the performance of personalized models (LM).
>
> Fed3+2p + FedDyn: The global model (GM) performance slightly improves, but local model (LM) performance decreases. The likely reason is that FedDyn uses a regularization term to encourage each client’s local model to stay close to the global model. While this enhances the consistency of the global model, improving its performance, it restricts the local model’s capacity to make client-specific adjustments, thereby reducing local performance.

---

> ### Author Response · Authors · 2024-11-21
> **Rebuttal by Authors**
>
> Q4: Can the authors clarify the statement "we turn off the Type-B coordinator in the second phase but retained the management functionality of the Type-A coordinator". Does this mean there is no second phase in this ablation? Also, additional experiments with random Type-B assignment would be informative.
>
> A4: Thank you for your question. "Disabling Type-B coordinators while retaining the management functionality of Type-A coordinators in the second phase" does not imply the absence of a second phase in this ablation study. Under this setup, all clients train independently in the second phase, and clients with similar data distributions no longer exchange information.
>
> To further illustrate the role of Type-B coordinators, we conducted additional experiments. These experiments followed the same setup as the hyperparameter experiments in the appendix, with a non-IID setting of Dir(0.1). Here, M represents the number of Type-B coordinators, R represents random selection, and NR represents selection according to the method used in Fed3+2p. The comparison focuses on the average test accuracy of local models. The results are as follows:
>
> | M  | 3    | 5    | 7    | 10   | 12   | 14   | 16   | 18   |
> |----|------|------|------|------|------|------|------|------|
> | R  | 90.4 | 90.4 | 90.6 | 91.3 | 91.5 | 92.2 | 92.9 | 93.5 |
> | NR | 92.9 | 93.6 | 93.5 | 93.6 | 93.7 | 94.1 | 93.8 | 93.6 |
>
> The results indicate that random selection significantly reduces the performance of local models. This demonstrates that the design of Type-B coordinators alleviates the overfitting problem for clients with smaller data volumes, thereby improving the local model performance.

---

> > ### Comment · Reviewer_RdTV · 2024-11-25
> > **Responce to rebuttal**
> >
> > I reviewed the authors' additional experiments and appreciated their efforts. Despite all this, I am still concerned about the technical novelty and what this technique can bring to the overall federated learning community. As a result, I keep my score for the paper.

---

> > > ### Author Response · Authors · 2024-11-26
> > > **Rebuttal by Authors**
> > >
> > > W1: The novelty of the approach is not sufficiently articulated. It's unclear whether: (a) The use of coordinators for client grouping is novel. Have similar dataset/client splitting approaches been tried before? (b) The two-phase training scheme is not novel as it is frequently seen in personalized federated learning (pFL). In terms of progressive training, the idea has been explored in "FedBug: A Bottom-Up Gradual Unfreezing Framework for Federated Learning"), and a direct comparison would be helpful to better understand the novelty.
> > >
> > > Q1(a): Thank you for highlighting the coordinators as a key innovation. However, we would like to clarify that, in addition to the coordinators, the filter is also an important innovation in our approach.
> > >
> > > Our method features the following innovations: the neural network architecture is divided into three parts—feature extractor, filter, and classification head—and we adopt a two-stage training strategy with two types of coordinators managing the clients:
> > >
> > > Stage 1: Training the global feature extractor
> > > To train the shared feature extractor for all clients under conditions that approximate the global data distribution, we introduce Class-A coordinators. These coordinators divide all clients into several groups, ensuring that their joint data distribution is as close as possible to the global data distribution, and update the feature extractor’s parameters at both the coordinator and central server.
> > >
> > > Stage 2: Training the local filter
> > > To train the shared filter for some clients under similar data distributions, we propose Class-B coordinators. These coordinators cluster clients with similar data distributions and update the filter’s parameters at the corresponding coordinators.
> > >
> > > To the best of our knowledge, there is no existing work that exactly matches our design.
> > >
> > > Q1(b): It needs to be clarified that the two-stage training scheme itself is not our primary innovation, and there is a fundamental difference between FedBug and our method, Fed3+2p.
> > >
> > > FedBug uses a method of freezing a single layer's parameters, which makes the direction of local updates more consistent with global updates, thus training a better-performing global model. On the other hand, our method, Fed3+2p, divides all clients into several groups using Class-A coordinators, ensuring that their joint data distribution is as close as possible to the global data distribution, which makes the direction of local updates more consistent with global updates, thus training a better-performing feature extractor.
> > >
> > > Furthermore, Fed3+2p is a federated learning approach that simultaneously achieves excellent performance for both global and local models, while FedBug only considers the performance of the global model and does not take into account the performance of local models.

---

### Official Review · Reviewer_VjZ8 · 2024-11-01

**Soundness:** 2
**Presentation:** 2
**Contribution:** 2
**Rating:** 3
**Confidence:** 4

**Summary:**

This paper aims to improve both global federated learning and personalized feature learning. It first decomposes the model into three parts: a feature extractor, a filter, and classification heads. Then, the clients are divided into different groups according to their data distribution. The training process employs two types of coordinators within a two-phase training strategy to train different parts of models. Experiments indicate that the proposed Fed3+2p method surpasses existing state-of-the-art approaches in both global and local performance.

**Strengths:**

1. Propose a new framework to enhance both global and local FL performance.
2. Experiments indicate that the proposed Fed3+2p method surpasses existing state-of-the-art approaches in both global and local performance.

**Weaknesses:**

**Method:**

1. The proposed filter seems to be the key innovation in this paper. However, the ablation experiments presented in Table 2 indicate that removing the filter results in only minor performance declines of 0.3, 0.4, and 0.1 on FashionMNIST, CIFAR-10, and CIFAR-100, respectively. This suggests that the filter may offer limited utility.
2. This method involves partitioning clients into distinct groups. I hypothesize that this requires clients to upload their data distributions to the server, which presents a potential privacy risk.

**Experiment:**

3. Could the author provide a comparison with FedETF [1]? This method also aims to enhance both personal and global performance.
4. The experiments are conducted on small ConvNet architectures consisting of 2 or 3 convolutional layers. I am curious about the performance on larger networks, such as ResNet.

**Writing:**

5. The paper's writing requires improvement. Additionally, the notation is inaccurate. For instance, high-dimensional tensors, such as data $x$,  should be bolded as $\mathbf{x}$ in eq. (1).
6. In Section 2.1, citations should be included when discussing methods related to categories.
7. Regarding the methods, it is advisable to include an algorithm.

[1] Li, Zexi, et al. "No fear of classifier biases: Neural collapse inspired federated learning with synthetic and fixed classifier." Proceedings of the IEEE/CVF International Conference on Computer Vision. 2023.

**Questions:**

1. The proposed filter seems the key innovation in this paper. Could the authors provide a more detailed explanation of the filter's motivation?
2. The filter's structure is unclear. Could the authors elaborate on its design and whether its implementation incurs significant computational overhead?
3. Could the author provide a comparison with FedETF [1]? This method also seeks to enhance both personal and global performance.
4. The experiments are conducted on small ConvNet architectures consisting of 2 or 3 convolutional layers. I am curious about the performance on larger networks, such as ResNet.

---

> ### Author Response · Authors · 2024-11-21
> **Rebuttal by Authors**
>
> Thank you for your valuable feedback. The following is our responses to the questions you raised.
>
> Q1: The proposed filter seems the key innovation in this paper. Could the authors provide a more detailed explanation of the filter's motivation?
>
> A1: We appreciate your interest in the filter as a key innovation. However, we would like to clarify that another significant innovation in our approach is the coordinator. The motivation behind the filter is to prevent clients with limited data from overfitting. To achieve this, the filter’s design learns from the knowledge of other clients whose data distributions are similar to those of the data-scarce clients. In fact, our ablation experiments show that when the filter is removed, the test accuracy of clients with limited data significantly drops compared to when the filter is present.
>
> Q2: The filter's structure is unclear. Could the authors elaborate on its design and whether its implementation incurs significant computational overhead?
>
> A2: In our experiments, we considered using either the last convolutional layer or the first fully connected layer as the filter. The experimental results showed that the latter performed better, which is why we present the results using the first fully connected layer as the filter. The computational overhead introduced by this implementation is minimal and mainly pertains to the client grouping process, which is negligible in terms of impact.
>
> Q3&4: Could the author provide a comparison with FedETF [1]? This method also seeks to enhance both personal and global performance. The experiments are conducted on small ConvNet architectures consisting of 2 or 3 convolutional layers. I am curious about the performance on larger networks, such as ResNet.
>
> A3&4: To better demonstrate the superiority of our method, we conducted additional experiments under the same settings as the paper "No Fear of Classifier Biases: Neural Collapse Inspired Federated Learning with Synthetic and Fixed Classifier", comparing Fed3+2p with FedETF. These experiments were conducted using the complex network ResNet20, and the results are as follows:
>
> | Dataset  | |   | |   | Cifar10 |   | |   ||   ||  | Cifar100|   |  |  |
> |----------|-|-----------|-|------------|-|------------|-|------------|-|------------|-|------------|-|------------|-|------------|
> |   Non-IID       || |Dir(0.1) |         | | |Dir(0.05) |         ||   |  Dir(0.1)     | |  || Dir(0.05)        ||          |
> |  Model Type       || GM | |     LM    || GM  | |  LM       | |GM | |LM  | |GM |   |  LM     |
> |  FedETF        || 59.56      || 87.89      | |56.08      | |92.62      || 26.24      | |52.86      | |24.17      | |60.68      |
> |  Fed3+2p      | |87.25      | |92.35      | |85.74      | |94.21      | |57.12      | |70.90      | |54.77      | |78.51      |
>
> The results clearly indicate that our method outperforms FedETF in both global model (GM) and local model (LM) performance. Furthermore, Fed3+2p exhibits excellent performance even on complex networks, further validating its superiority and generalizability.

---

### Official Review · Reviewer_V2py · 2024-11-05

**Soundness:** 2
**Presentation:** 1
**Contribution:** 1
**Rating:** 3
**Confidence:** 5

**Summary:**

The paper proposes a new federated learning framework called Fed3+2p, which aims to address the impact of non-iid data on global and local performance, as well as the overfitting issues that small-data clients may encounter. The framework divides the client neural network into three parts: feature extractor, filter, and classifier head, and trains these parts using a two-stage strategy with two types of coordinators. Experimental results show that Fed3+2p outperforms existing methods on the FMNIST and CIFAR-10/100 datasets.

**Strengths:**

The design idea of dividing the client neural network into three parts and training them using a two-stage strategy has some practical application value.

**Weaknesses:**

1. The setup of the paper is not clear. In general, in federated learning, training should protect the privacy of each client, and the clients selected for training in each round should be random. The method proposed in this paper requires obtaining the class distribution of each client, which to some extent violates the principle of privacy protection in federated learning. Additionally, controlling the clients participating in training in each round contradicts the standard setup of federated learning.
2. The method proposed in the paper is too simple, and I did not find any unique or innovative aspects.
3. The notation in the paper is confusing and the descriptions are unclear, making it difficult to read. For example, C is used to represent both coordinators and categories.
4. The experiments are insufficient:
-  The authors seem to have shown only one set of experimental results under a single setting, which has a high degree of randomness and is not sufficiently comprehensive.
- Important settings such as how many clients the data was divided into and how many clients were randomly selected for each communication are missing from the paper. Additionally, crucial details like how the number of coordinators should be determined are also absent.
- Many papers[1][2][3] that consider class imbalance in federated learning are not included in the comparison methods.

[1] On Bridging Generic and Personalized Federated Learning for Image Classification.

[2] No Fear of Classifier Biases: Neural Collapse Inspired Federated Learning with Synthetic and Fixed Classifier.

[3] Aligning model outputs for class imbalanced non-IID federated learning.

**Questions:**

Refer to the weakness.

---

> ### Author Response · Authors · 2024-11-21
> **Rebuttal by Authors**
>
> Thank you for your valuable feedback. The following is our responses to the questions you raised.
>
> Q1: The setup of the paper is not clear. In general, in federated learning, training should protect the privacy of each client, and the clients selected for training in each round should be random. The method proposed in this paper requires obtaining the class distribution of each client, which to some extent violates the principle of privacy protection in federated learning. Additionally, controlling the clients participating in training in each round contradicts the standard setup of federated learning.
>
> A1: In the Fed3+2p framework, client classification does rely on label distribution information. We acknowledge that label distribution could potentially be exploited in attacks. However, to date, we have not encountered any concrete methods that can infer original data solely from label distribution information. Furthermore, similar approaches using label distribution information can be found in the following references:
>
> [1] Tao Sheng, Chengchao Shen, Yuan Liu, Yeyu Ou, Zhe Qu, and Jianxin Wang. Modeling global distribution for federated learning with label distribution skew, 2022. URL: https://arxiv.org/abs/2212.08883
>
> [2] Jun Luo and Shandong Wu. FedSLD: Federated learning with shared label distribution for medical image classification, 2021. URL: https://arxiv.org/abs/2110.08378.
>
> [3] A. Khalil et al., "Label-Aware Aggregation for Improved Federated Learning," 2023 Eighth International Conference on Fog and Mobile Edge Computing (FMEC), Tartu, Estonia, 2023, pp. 216-223, doi: 10.1109/FMEC59375.2023.10306055.
>
> [4] Zhu Z, Hong J, Zhou J. Data-free knowledge distillation for heterogeneous federated learning[C]//International conference on machine learning. PMLR, 2021: 12878-12889.
>
> Regarding the selection of training clients, we did not impose strict controls on which clients participate in each training round. For the FMNIST dataset, 20% of clients under each A/B coordinator were randomly selected for training in each round. For the CIFAR-10/100 datasets, 40% of clients per A/B coordinator were randomly chosen. In fact, many federated learning methods adopt similar approaches, as demonstrated in the following reference:
>
> [5] Md Sirajul Islam, Simin Javaherian, Fei Xu, Xu Yuan, Li Chen, and Nian-Feng Tzeng. FedClust: Tackling data heterogeneity in federated learning through weight-driven client clustering, 2024. URL: https://arxiv.org/abs/2407.07124.
>
> Q2: The method proposed in the paper is too simple, and I did not find any unique or innovative aspects.
>
> A2: We believe that the superiority of a method should be evaluated based on its performance rather than its complexity. Our experimental results demonstrate that our approach achieves performance comparable to, or even surpassing, several existing methods on commonly used datasets. These findings validate the effectiveness of our approach. Additionally, compared to more complex methods, the simplified design of our framework enhances its interpretability and feasibility in implementation.
>
> Our method introduces a novel approach by dividing the neural network architecture into three components: a feature extractor, a filter, and classification heads, and employing a two-phase training strategy with two types of coordinators managing the clients:
>
> Phase 1: Training the global feature extractor
> To train the shared feature extractor for all clients under conditions approximating the global data distribution, we utilize Type-A coordinators. These coordinators partition all clients into several groups such that their combined data distributions resemble the global distribution. The parameters of the feature extractor are then updated on both the coordinators and the central server.
>
> Phase 2: Training the local filter
> To train the shared filter among clients with similar data distributions, we propose Type-B coordinators, which cluster clients with comparable data distributions and update the filter parameters on their corresponding coordinator.
>
> To the best of our knowledge, no existing work employs a design identical to ours.

---

> ### Author Response · Authors · 2024-11-21
> **Rebuttal by Authors**
>
> Q3: The notation in the paper is confusing and the descriptions are unclear, making it difficult to read. For example, C is used to represent both coordinators and categories.
>
> A3: Thank you for your valuable feedback. We acknowledge that the notation may cause confusion, particularly with "C" being used to represent both coordinators and categories. To improve clarity and readability, we will redefine the notations and ensure that each symbol has a clear and consistent meaning throughout the paper. We will make these adjustments in the revised version to better communicate our method and ideas, avoiding any ambiguity and enhancing the overall readability of the paper.
>
> Q4.1: The authors seem to have shown only one set of experimental results under a single setting, which has a high degree of randomness and is not sufficiently comprehensive.
>
> A4.1: Dir(0.1) and Dir(0.3) are commonly used non-IID settings in federated learning experiments. As the paper "On Bridging Generic and Personalized Federated Learning for Image Classification" is the main baseline for comparison in our work, we followed its use of these settings. Due to space limitations, we included the results for Dir(0.1) in the main text, while the results for Dir(0.3) are provided in the appendix.
>
> It is worth noting that the conclusions and analysis under both settings are nearly identical. To ensure the stability of our results, we used different random seeds and conducted 5 independent runs to compute the average. This approach effectively controls for the impact of randomness, ensuring the reliability of our findings.

---

> ### Author Response · Authors · 2024-11-21
> **Rebuttal by Authors**
>
> Q4.2&4.3:
>
> Important settings such as how many clients the data was divided into and how many clients were randomly selected for each communication are missing from the paper. Additionally, crucial details like how the number of coordinators should be determined are also absent.
>
> Many papers[1][2][3] that consider class imbalance in federated learning are not included in the comparison methods.
>
> [1] On Bridging Generic and Personalized Federated Learning for Image Classification.
>
> [2] No Fear of Classifier Biases: Neural Collapse Inspired Federated Learning with Synthetic and Fixed Classifier.
>
> [3] Aligning model outputs for class imbalanced non-IID federated learning.
>
> A4.2&4.3:
>
> Regarding the experimental setup, FedRod from [1] is a primary baseline for comparison with our method. To ensure fairness, we strictly followed the experimental settings of [1], including the number of clients and the randomly selected clients per round for communication. Therefore, we are quite puzzled by your mention that "[1] was not included in the comparison methods."
>
> Regarding your question on "how the number of coordinators is determined," due to space constraints, we only presented the results with the optimal number of coordinators in the main text. The complete hyperparameter experiments, including the impact of changing the number of coordinators, are detailed in the appendix for your reference.
>
> Moreover, to further demonstrate the superiority of our method, we conducted additional experiments under the same settings as [2], comparing Fed3+2p with FedETF. The results are as follows:
>
> | Dataset  | |   | |   | Cifar10 |   | |   ||   ||  | Cifar100|   |  |  |
> |----------|-|-----------|-|------------|-|------------|-|------------|-|------------|-|------------|-|------------|-|------------|
> |   Non-IID       || |Dir(0.1) |         | | |Dir(0.05) |         ||   |  Dir(0.1)     | |  || Dir(0.05)        ||          |
> |  Model Type       || GM | |     LM    || GM  | |  LM       | |GM | |LM  | |GM |   |  LM     |
> |  FedETF        || 59.56      || 87.89      | |56.08      | |92.62      || 26.24      | |52.86      | |24.17      | |60.68      |
> |  Fed3+2p      | |87.25      | |92.35      | |85.74      | |94.21      | |57.12      | |70.90      | |54.77      | |78.51      |
>
> The results clearly show that our method outperforms FedETF in both global (GM) and local model (LM) performance.
>
> Finally, regarding [3], it is important to clarify that Fed3+2p is an independent federated learning method designed to achieve both superior global model (GM) and local model (LM) performance. In contrast, [3] is not a standalone federated learning method but rather a combination of two techniques: probability correction loss (Pc) and pre-defined prototypes (Pp). Moreover, the work in [3] focuses solely on global model performance and does not consider local model performance. Given its limited relevance to our study, we did not include [3] as a primary comparison method in our paper.
>
> Additionally, to further explore the compatibility of the techniques proposed in [3] with our method, we conducted additional experiments under the same settings as described in our main paper. Specifically, we combined our Fed3+2p approach with Pc and Pp to examine whether these techniques could further improve performance. The results are summarized below:
>
> |     Dataset     | |          | |  Cifar10    | |     | |     | |  Cifar100      | | |
> |------|-|-------|-|-------|-|--------|-|-------|-|-------|-|-------|
> |     Model Type     | |          |GM |          | | LM    | |     |GM  |       || LM |
> |     Method/Test Set     ||      G-Test     | |   P-Test      ||  P-Test    | |  G-Test   ||  P-Test       || P-Test |
> |     Fed3+2p + Pc&Pp     |  |    80.7		   | |    80.9	     | | 91.7	    |  |38.9	    | |    38.7    | |66.5    |
> |      Fed3+2p    |  |   81.4    | |   83.7	      ||   94.1	   | |  45.9   |  |  45.3	      || 72.6    |
>
> The results indicate that integrating Pc and Pp into the Fed3+2p framework resulted in a decline in performance. We attribute this phenomenon to potential conflicts between the mechanisms of Pc and Pp and the personalization strategy of Fed3+2p:
>
> Pc, which optimizes GM training through probability adjustment, may overfit the global model and thereby compromise LM performance in the Fed3+2p framework.
>
> Pp, which constrains the feature space with predefined prototypes, could limit the P-Head's ability to adaptively adjust features, thereby reducing personalized performance.
>
> We hope this analysis provides a clear explanation and appreciate your feedback for further discussion.

---

> > ### Comment · Reviewer_V2py · 2024-11-26
> >
> > I have carefully read the authors' rebuttal and I believe this paper does not meet the quality standards for publication at ICLR. I have decided to maintain my score based on the following reasons:
> > 1. As of the time I submit this comment, the authors do not seem to have uploaded a revised version of the paper. I am unable to assess the improvements in readability that the authors claim to have made.
> > 2. I do not agree with the authors' statement that "we have not encountered any concrete methods that can infer original data solely from label distribution information." The grouping in the paper is based on calculating the KL divergence between the class distributions of data on different clients. This inevitably leads to the leakage of the class distribution information of client data, which seriously violates the principle of federated learning.
> > 3. The authors seem to have omitted some important settings in the experimental section of the paper. For example, in the rebuttal, they mention conducting experiments repeated five times and selecting 20% randomly in each round. I did not find any related descriptions in the paper. Additionally, in the experimental results, only the mean is reported without variance, and I still have doubts about the experimental results. At the same time, the authors' grouping is based on different clients, but there is a serious lack of relevant experimental studies on the grouping situation. For example, if the KL divergence between two clients is the same (considering an extreme case where each client has only one class of samples), how should grouping be done? What impact would it have if most clients are grouped into one group, while other groups have only one client? This paper is incomplete.
> > 4. The work presented in this paper is largely a straightforward combination of established techniques, without offering any significant advancements or novel insights. The lack of innovation limits its impact on the field.

---

### Note · Authors · 2024-11-26

I have read and agree with the venue's withdrawal policy on behalf of myself and my co-authors.